# Excitation of medium spiny neurons by 'inhibitory' ultrapotent chemogenetics via shifts in chloride reversal potential

Stephanie C Gantz[1]*, Maria M Ortiz[2], Andrew J Belilos[3], Khaled Moussawi[3,4]*

[1]Department of Molecular Physiology and Biophysics, University of Iowa, Iowa City, United States; [2]Biological and Biomedical Neuroscience Program, University of North Carolina, Chapel Hill, United States; [3]National Institute on Drug Abuse, Baltimore, United States; [4]Department of Psychiatry, School of Medicine, University of Pittsburgh, Pittsburgh, United States

**Abstract** Ultrapotent chemogenetics, including the chloride-permeable inhibitory PSAM[4]-GlyR receptor, were recently proposed as a powerful strategy to selectively control neuronal activity in awake, behaving animals. We aimed to validate the inhibitory function of PSAM[4]-GlyR in dopamine D1 receptor-expressing medium spiny neurons (D1-MSNs) in the ventral striatum. Activation of PSAM[4]-GlyR with the uPSEM[792] ligand enhanced rather than suppressed the activity of D1-MSNs in vivo as indicated by increased c-fos expression in D1-MSNs and in vitro as indicated by cell-attached recordings from D1-MSNs in mouse brain slices. Whole-cell recordings showed that activation of PSAM[4]-GlyR depolarized D1-MSNs, attenuated GABAergic inhibition, and shifted the reversal potential of PSAM[4]-GlyR current to more depolarized potentials, perpetuating the depolarizing effect of receptor activation. These data show that 'inhibitory' PSAM[4]-GlyR chemogenetics may activate certain cell types and highlight the pitfalls of utilizing chloride conductances to inhibit neurons.

*For correspondence:
stephanie-gantz@uiowa.edu (SCG);
moussawi.khaled@gmail.com (KM)

Competing interests: The authors declare that no competing interests exist.

## Introduction

Novel research tools like opto- and chemogenetics have been instrumental in dissecting brain circuits and understanding their relevance to normal and maladaptive behaviors. However, like any new technology, these tools have inherent limitations that could confound data interpretation.

The leading chemogenetic approach, Designer Receptors Exclusively Activated by Designer Drugs (DREADDs), acts through engineered G protein-coupled receptors, which are selectively activated by clozapine N-oxide (CNO) (*Roth, 2016*). However, DREADDs may disrupt signaling from endogenous G protein-coupled receptors when expressed at high levels (*Saloman et al., 2016*), and CNO has been found to be metabolized peripherally to clozapine, which binds to many receptors in the brain, potentially resulting in off-target effects (*Gomez et al., 2017*; *Manvich et al., 2018*). A more recent chemogenetic approach, Pharmacologically Selective Actuator/Effector Module (PSAM/PSEM), acts through engineered ligand-gated ion channels that are activated by PSEM agonists (*Magnus et al., 2011*; *Magnus et al., 2019*). The inhibitory PSAM[4]-GlyR is a chimeric protein of a modified α7 nicotinic acetylcholine receptor ligand binding domain fused to the chloride-permeable glycine receptor ion pore domain (GlyR). After intracranial injection and expression of a virus encoding PSAM[4]-GlyR, application of ultrapotent agonist (e.g., uPSEM[792]) activates PSAM[4]-GlyR, which in principle selectively inhibits PSAM[4]-GlyR-expressing neurons (*Magnus et al., 2019*).

In this study, we aimed to validate the inhibitory function of PSAM[4]-GlyR in dopamine D1 receptor-expressing medium spiny neurons (D1-MSNs) in the ventral striatum. Intraperitoneal (i.p.) injection of uPSEM[792] enhanced the expression of the immediate early gene c-fos in PSAM[4]-GlyR-

expressing D1-MSNs, indicative of in vivo activation rather than inhibition of these neurons. Using whole-cell recordings in acute brain slices, we found that activation of PSAM[4]-GlyR with uPSEM[792] decreased membrane resistance, induced an inward current in voltage-clamp, and depolarized the membrane potential in current-clamp, sometimes resulting in depolarization block. The majority of neurons maintained action potential firing to somatic current injection. Cell-attached recordings at subthreshold depolarized potentials also showed that activation of PSAM[4]-GlyR induced firing of action potentials in D1-MSNs. Further, we found that chloride influx via PSAM[4]-GlyR activation shifted its reversal potential to more positive values further exacerbating the depolarizing effect and attenuated GABAergic inhibition of D1-MSNs.

## Results

### Selective expression of PSAM[4]-GlyR in D1-MSNs

To first validate the selective expression of PSAM[4]-GlyR in ventral striatum D1-MSNs, prodynorphin-Cre mice were crossed with a Cre-reporter (Ai9) mouse line, resulting in labeling of neurons containing Cre-recombinase (i.e., D1-MSNs) with the fluorescent reporter tdTomato (*Al-Hasani et al., 2015*; *Gerfen et al., 1990*; *Krashes et al., 2014*). These mice received bilateral stereotaxic microinjections of AAV-syn-flex-PSAM[4]-GlyR-IRES-eGFP. PSAM[4]-GlyR expression was restricted to tdTomato[+] neurons which confirmed selective Cre-dependent expression of PSAM[4]-GlyR in D1-MSNs (*Figure 1A*).

### PSAM[4]-GlyR enhances rather than suppresses c-fos expression in D1-MSNs in vivo

The immediate early gene c-fos is commonly used as a surrogate marker for neuronal activation (*Chung, 2015*; *Cruz et al., 2015*; *Sheng and Greenberg, 1990*; *Hunt et al., 1987*). Given the proposed role of PSAM[4]-GlyR activation in silencing neurons (*Magnus et al., 2011*; *Magnus et al., 2019*), we initially hypothesized that activating PSAM[4]-GlyR in vivo will suppress D1-MSNs activity and reduce c-fos expression. To avoid a potential floor effect, we treated mice with fentanyl since opioid exposure has been shown to increase the expression of c-fos and other immediate early genes in striatal neurons including D1-MSNs (*Enoksson et al., 2012*; *Lobo et al., 2013*; *Chang et al., 1988*; *Liu et al., 1994*). Prodynorphin-Cre mice received bilateral injections of either AAV-syn-flex-PSAM[4]-GlyR-IRES-eGFP (PSAM[4]-GlyR) or AAV-EF1α-DIO-eYFP. Then, after 4–6 weeks, mice were injected with uPSEM[792] (i.p., 3 mg/kg) or saline, followed 30 min later by fentanyl (i.p., 0.2 mg/kg) or saline (*Figure 1B*). Mice were perfused and their brains collected for c-fos immunostaining 90 min later (120 min after the first i.p. injection). Transduced neurons were identified by expression of the fluorescent reporter (eGFP or eYFP, green, *Figure 1C*). Contrary to expectation, activation of PSAM[4]-GlyR with i.p. uPSEM[792] increased c-fos expression in D1-MSNs expressing PSAM[4]-GlyR (*Figure 1D*). Two-way ANOVA showed a significant effect of PSAM[4]-GlyR activation on c-fos expression ($F_{1, 34} = 30.55$, p<0.0001). The experimental groups included 8–12 animals/group (control + saline: n = 12; PSAM[4]-GlyR + saline: n = 8; control + fentanyl: n = 9; PSAM[4]-GlyR + fentanyl: n = 9). These results support the rejection of the initial hypothesis that PSAM[4]-GlyR activation reduces c-fos expression in D1-MSNs. There was no difference in c-fos expression in non-transduced cells (two-way ANOVA; PSAM[4]-GlyR activation effect: $F_{1, 34} = 0.82$, p=0.37; *Figure 1E*). These results show that activation of PSAM[4]-GlyR increased c-fos expression in transduced cells and therefore activated rather than inhibited D1-MSNs in vivo.

### PSAM[4]-GlyR activation depolarizes D1-MSNs

To test the effects of PSAM[4]-GlyR on electrophysiological properties of D1-MSNs, PSAM[4]-GlyR-encoding virus was injected into the ventral striatum of prodynorphin-Cre mice. After 4–6 weeks, whole-cell patch-clamp recordings ($V_{hold}$ −88 mV, corrected for junction potential) were made from eGFP[+] neurons in acute mouse brain slices. The internal solution in these recordings was potassium-based and contained 12.8 mM Cl[−] to mimic the natural reversal potential of chloride in MSNs ($E_{Cl}$ ~ −62 mV) (*Misgeld et al., 1982*; *Bracci and Panzeri, 2006*). PSAM[4]-GlyR was activated by application of the ligand uPSEM[792]. To control for any off-target effects of uPSEM[792], recordings were also made from control neurons (neighboring eGFP[−] neurons and eYFP[+] neurons from mice that received bilateral injections of AAV-EF1α-DIO-eYFP). In control neurons, 10 or 50 nM uPSEM[792] had

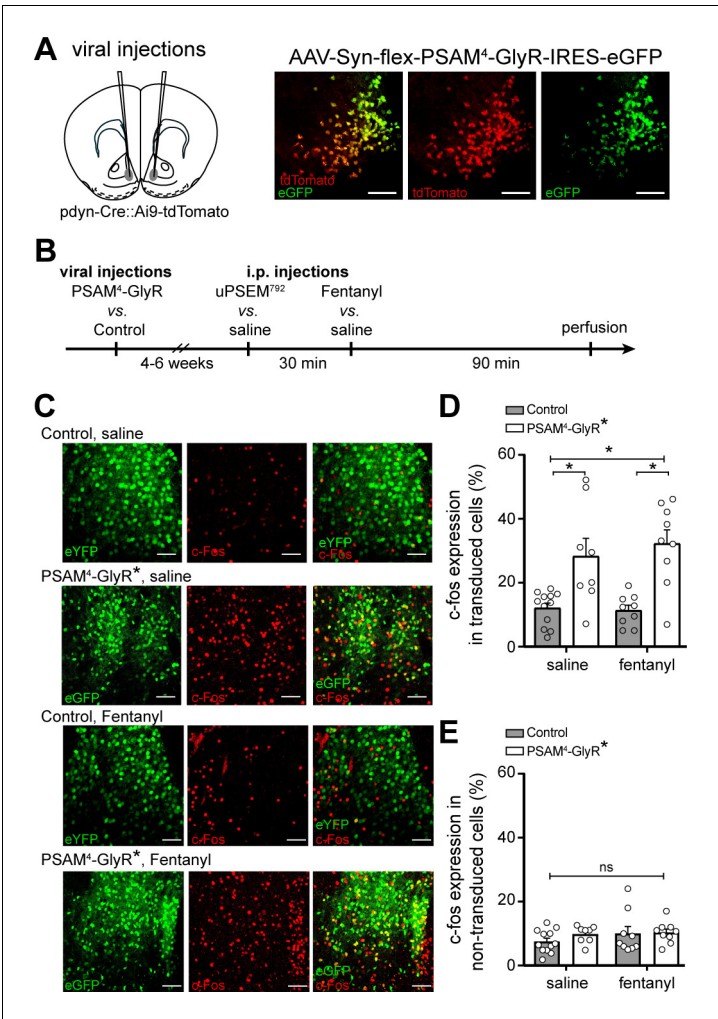

**Figure 1.** In vivo activation of PSAM[4]-GlyR enhances c-fos expression in D1-MSNs. (**A**) Left, cartoon of viral injection site in the ventral striatum. Right, representative confocal images of Cre-dependent PSAM[4]-GlyR expression (eGFP) in Cre-expressing D1-MSNs (tdTomato). (**B**) Experimental design for measuring c-fos expression after activation of PSAM[4]-GlyR. (**C**) Representative maximum intensity Z-stack confocal images of c-fos immunostaining (red) in D1-MSNs with or without activation of PSAM[4]-GlyR, after saline *vs.* fentanyl injection (0.2 mg/kg, i.p.). PSAM[4]-GlyR* refers to PSAM[4]-GlyR activated with uPSEM[792] (3 mg/kg, i.p.). The 'Control' group includes pooled data from mice with (control virus + uPSEM[792] 1st i.p. injection) and (PSAM[4]-GlyR virus + saline 1st i.p. injection). Scale bars: 50 μm. (**D**) Activation of PSAM[4]-GlyR with uPSEM[792] increased c-fos expression in PSAM[4]-GlyR expressing D1-MSNs compared to control (two-way ANOVA with Sidak's multiple comparisons test). Data are presented as % colocalization of c-fos and eGFP/eYFP[+] in transduced D1-MSNs. (**E**) Activation of PSAM[4]-GlyR did not increase c-fos expression in non-transduced cells (calculated as the number of c-fos expressing non-transduced cells/total number of non-transduced cells). Data represent mean ± SEM. * indicates statistical significance, ns denotes not significant.

The online version of this article includes the following source data for figure 1:

**Source data 1.** PSAM4-GlyR enhances c-fos expression in D1-MSNs in-vivo (source data).

no effect on basal whole-cell current (10 nM: $-2.8 \pm 7.7$ pA, $n = 6$, p=0.73, paired $t$-test; 50 nM: $-9.5 \pm 9.4$ pA, $n = 14$, p=0.24; *Figure 2A*). In PSAM[4]-GlyR[+] neurons, 10 nM uPSEM[792] produced an inward current ($-43.0 \pm 10.1$ pA, $n = 21$, p=0.0004, paired $t$-test; *Figure 2A and B*), consistent with the original report on PSAM[4]-GlyRs (*Magnus et al., 2019*). Higher concentrations of uPSEM[792] produced substantially larger inward currents (50 nM: $-275.5 \pm 57.1$ pA, $n = 18$, p=0.0002, paired $t$-test; 100 nM: $-410.1 \pm 137.8$ pA, $n = 6$, p=0.031; *Figure 2A and B*). To assay membrane resistance, the instantaneous change in current in response to a 10 mV step (V$_{hold}$ $-88$ to $-78$ mV) was

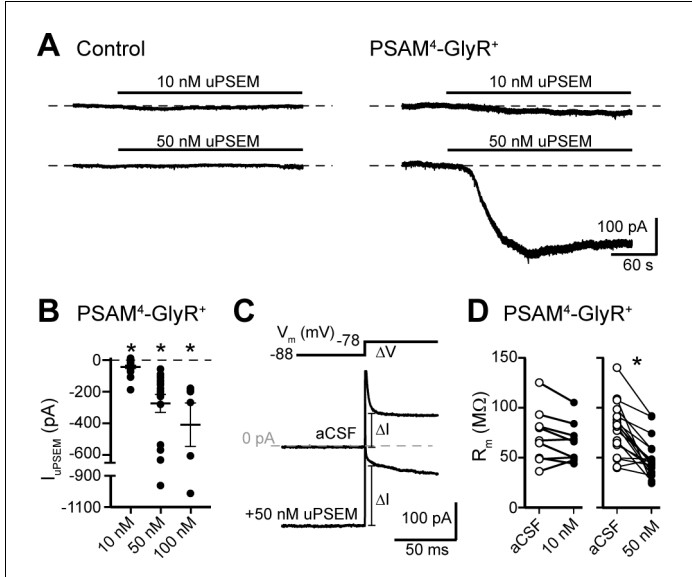

**Figure 2.** Activation of PSAM⁴-GlyR in D1-MSNs produces an inward current. (**A**) Representative traces of whole-cell voltage-clamp recordings (V_hold −88 mV) demonstrate no effect of uPSEM⁷⁹² (10 or 50 nM) in control neurons. In PSAM⁴-GlyR⁺ neurons, 10 or 50 nM uPSEM⁷⁹² produced an inward current. Dashed line is baseline whole-cell current for ease of visualization. (**B**) Plot of the magnitude of the inward current produced by 10, 50, or 100 nM uPSEM⁷⁹² in PSAM⁴-GlyR⁺ neurons. Line and error bars represent means ± SEM. (**C**) To measure membrane resistance ($R_m$), a 10 mV voltage step (−88 to −78 mV) was made in aCSF and during the uPSEM⁷⁹²-induced inward current, and the instantaneous change in current following the capacitive transient was measured (ΔI). Representative traces are shown below the voltage step command. Dashed line is 0 pA. (**D**) In PSAM⁴-GlyR⁺ neurons, 50 nM uPSEM⁷⁹² significantly decreased $R_m$ (paired $t$-test, p<0.0001) while 10 nM showed a trend toward lower $R_m$ (paired $t$-test, p=0.08). * indicates statistical significance, ns denotes not significant.

The online version of this article includes the following source data for figure 2:

**Source data 1.** of PSAM4-GlyR depolarizes D1-MSNs (source data).

---

measured before and after uPSEM⁷⁹² application (**Figure 2C**). The basal membrane resistance of D1-MSNs was 78.6 ± 5.5 MΩ, consistent with a previous report using similar recording conditions (**Planert et al., 2013**), and was reduced by activation of PSAM⁴-GlyR (10 nM: 6.5 ± 3.3% decrease, $n$ = 9, p=0.08, paired $t$-test; 50 nM: 29.6 ± 5.3% decrease, $n$ = 18, p<0.0001; **Figure 2D**).

## PSAM⁴-GlyR does not silence D1-MSN action potential firing in whole-cell recordings

We next determined the effect of PSAM⁴-GlyR activation on neuronal excitability. Action potential firing was evoked by somatic current injections (50 pA, 2 s) during whole-cell current-clamp recordings from control and PSAM⁴-GlyR⁺ neurons. PSAM⁴-GlyR activation by uPSEM⁷⁹² significantly depolarized the membrane potential of PSAM⁴-GlyR⁺ neurons compared to control neurons (**Figure 3A and B**). Two-way ANOVA showed a significant effect of PSAM⁴-GlyR activation on membrane potential ($F_{1,\ 37}$ = 9.70, p=0.004), and a significant effect of uPSEM⁷⁹² concentration ($F_{1,\ 37}$ = 4.96, p=0.03). Relative to the resting membrane potential (−83.3 ± 0.7 mV), the depolarization induced by 10 and 50 nM uPSEM⁷⁹² was 6.4 ± 2.0 mV ($n$ = 14, p=0.007, paired $t$-test) and 20.9 ± 3.4 mV ($n$ = 16, p<0.0001), respectively.

In control neurons, uPSEM⁷⁹² (50 nM) did not change the number of evoked action potentials ($n$ = 4–7, ANOVA mixed-effects analysis; no significant group effect: $F_{1,\ 6}$ = 0.83, p=0.40, and no interaction: $F_{2.1,\ 8.73}$ = 1.30, p=0.32; **Figure 3B and F**) or firing frequency (no significant group effect: $F_{1,\ 6}$ = 0.29, p=0.61, and no interaction: $F_{2.97,\ 12.55}$ = 2.78, p=0.08; **Figure 3G**). In PSAM⁴-GlyR⁺ neurons, all neurons treated with 10 nM uPSEM⁷⁹² continued to fire action potentials with somatic current injections ($n$ = 14, **Figure 3C**). In 50 nM uPSEM⁷⁹², 10 of 16 neurons continued to

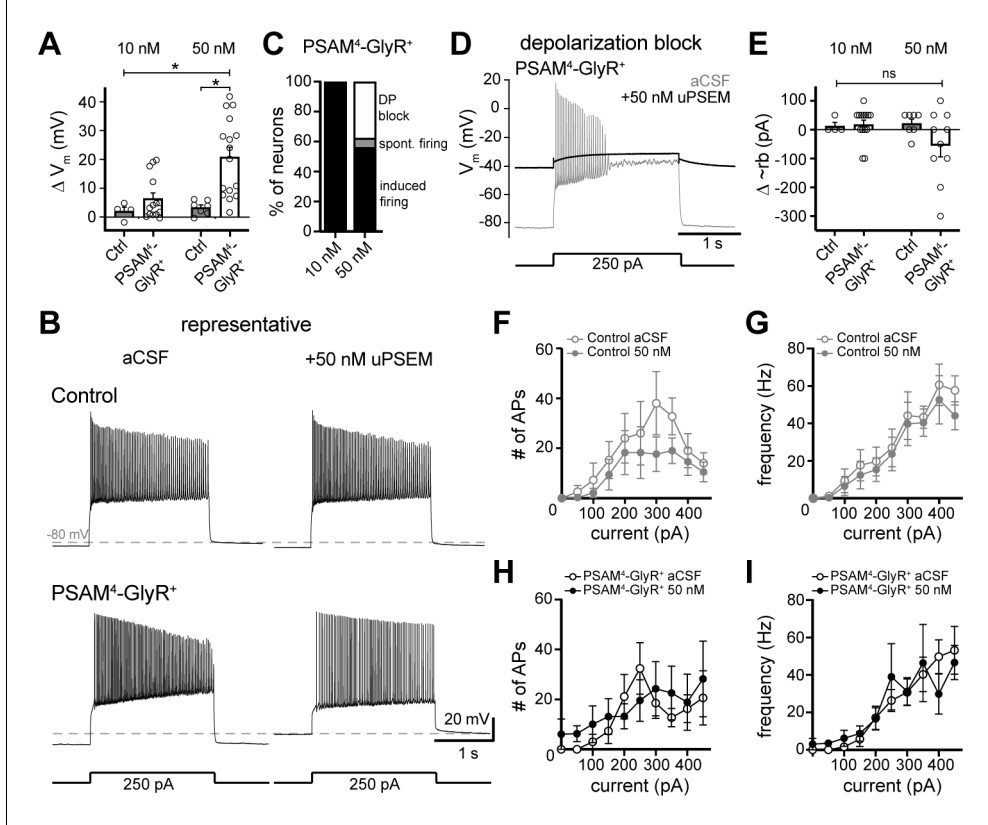

**Figure 3.** Activation of PSAM[4]-GlyR depolarizes D1-MSNs and does not inhibit firing. (**A**) In PSAM[4]-GlyR[+] neurons, activation of PSAM[4]-GlyR with uPSEM[792] (10 and 50 nM) significantly depolarized the membrane potential ($V_m$) compared to control (non-PSAM[4]-GlyR) neurons, measured in current-clamp (two-way ANOVA with Sidak's multiple comparisons test). (**B**) Representative traces of current-clamp recordings from a control (top) or PSAM[4]-GlyR[+] neuron (bottom). Action potential firing was evoked by somatic current injection (250 pA, 2 s) in aCSF (left) and 50 nM uPSEM[792] (right). Dashed line is −80 mV. (**C**) Distribution of firing response of PSAM[4]-GlyR[+] neurons after 10 or 50 nM uPSEM[792] application. In 10 nM uPSEM[792], 14/14 neurons continued to fire with current injection. In 50 nM, 9/16 neurons fired with current injection (induced firing), 1/16 fired spontaneously without a current injection (spont. firing), and 6/16 went into depolarization block (DP block). (**D**) Representative traces of current-clamp recordings from a PSAM[4]-GlyR[+] neuron that went into depolarization block with uPSEM[792]. Response to somatic current injection (250 pA, 2 s) in aCSF (gray) and after uPSEM[792] application (black). The membrane potential in uPSEM[792] was more depolarized than the potential where the neuron entered depolarization block in aCSF. Dashed line is −80 mV. (**E**) Plot of change in the minimum current needed to induce firing (approximate rheobase: ~rb) induced by uPSEM[792] (10 and 50 nM) in control (gray) and PSAM[4]-GlyR[+] (black) in cells that continued to fire (10/16). No significant change of ~rb was observed. (**F**) Plot of number of action potentials (APs) fired versus injected current in control neurons in aCSF (open) and in uPSEM[792] (closed). No significant difference was observed. (**G**) Plot of AP firing frequency versus injected current in control neurons in aCSF (open) and in uPSEM[792] (closed). No significant difference was observed. (**H**) Plot of number of APs fired versus injected current in PSAM[4]-GlyR[+] neurons in aCSF (open) and in uPSEM[792] (closed). No significant difference was observed. (**I**) Plot of AP firing frequency versus injected current in PSAM[4]-GlyR[+] neurons in aCSF (open) and in uPSEM[792] (closed). No significant difference was observed. Line and error bars represent means ± SEM, * indicates statistical significance, ns denotes not significant.

The online version of this article includes the following source data for figure 3:

**Source data 1.** PSAM4-GlyR does not inhibit firing of D1-MSNs (source data).

fire action potentials with current injection and 6 of 16 neurons were depolarized sufficiently to enter into depolarization block upon uPSEM[792] application (*Figure 3C and D*). The minimum positive current needed to evoke firing (approximation of rheobase) could not be determined in neurons that entered into depolarization block (6/16) upon activation of PSAM[4]-GlyR. When compared with neurons that continued to fire, neurons that entered into depolarization block with uPSEM[792] application

had similar resting membrane potential to other PSAM4-GlyR neurons ($-83.1 \pm 1.1$ mV *vs.* $-83.6 \pm 1.3$ mV), but had higher membrane resistance ($92.4 \pm 5.2$ mΩ *vs.* $65.8 \pm 6.7$ mΩ), indicating that depolarization block was not due to a generally depolarized membrane potential prior to activation of PSAM$^4$-GlyR. In neurons that continued to fire (10/16), there was no significant change in approximation of rheobase despite membrane depolarization with PSAM$^4$-GlyR activation, likely due to shunting (two-way ANOVA; PSAM$^4$-GlyR activation effect: $F_{1, 31} = 1.17$, p=0.29; uPSEM$^{792}$ concentration effect: $F_{1, 31} = 1.45$, p=0.24; *Figure 3E*). Moreover, activation of PSAM$^4$-GlyR with 50 nM UPSEM$^{792}$ did not change the number of action potentials ($n$ = 5–10, ANOVA mixed-effects analysis; no significant group effect: $F_{1, 9} = 0.20$, p=0.66, and no interaction: $F_{9, 49} = 0.96$, p=0.48; *Figure 3H*) or firing frequency (no significant group effect: $F_{1, 9} = 0.05$, p=0.82, and no interaction: $F_{9, 48} = 0.66$, p=0.48; *Figure 3I*).

## PSAM$^4$-GlyR induces action potential firing in D1-MSNs in cell-attached recordings

Whole-cell recording conditions affect intracellular chloride levels, which could confound data interpretation. Therefore, we sought to determine the effect of PSAM$^4$-GlyR activation on D1-MSN firing using loose cell-attached recording configuration, which does not perturb intracellular chloride concentrations. After obtaining a steady baseline, uPSEM$^{792}$ (100 nM) was applied for 5 min, followed by high potassium-containing CSF ($\sim$40 mM) to depolarize the neurons. High potassium results in rapid depolarization and firing by shifting the reversal potential of potassium to more positive values. Activation of PSAM$^4$-GlyR in D1-MSNs did not suppress high-potassium-induced firing of action potentials (*Figure 4A and B*). In fact, in one of five neurons, activation of PSAM$^4$-GlyR induced firing prior to high potassium application (not shown). Further, the frequency of high-potassium-induced firing was not different in uPSEM-treated control *vs.* PSAM$^4$-GlyR$^+$ neurons, suggesting the absence of a silencing effect by PSAM$^4$-GlyR activation.

The whole-cell recording data in *Figure 3A* suggest that PSAM$^4$-GlyR activation results in $\sim$20 mV depolarization of D1-MSNs, which is not sufficient to reach firing threshold ($\sim -43$ mV [Gertler et al., 2008]) from the resting membrane potential ($\sim -83$ mV). Therefore, to determine if the membrane depolarization induced by PSAM$^4$-GlyR activation is capable of triggering action potential firing when the membrane potential is closer to firing threshold, the cell-attached experiment was repeated after depolarizing the neurons to subthreshold potentials by increasing the extracellular potassium concentration from 4.5 mM to $\sim$11.5 mM (10–13 mM). Under these conditions, cell-attached recordings show that activation of PSAM$^4$-GlyR triggered firing of D1 MSNs (*Figure 4C and D*). In control cells, uPSEM$^{792}$ did not result in firing, unlike high potassium application (*Figure 4C and D*). A mixed-effects two-way ANOVA showed a significant interaction effect between PSAM$^4$-GlyR expression (+ *vs.* −) and recording condition (aCSF *vs.* uPSEM *vs.* high potassium) on the firing frequency ($F_{2, 13} = 12.40$, p=0.001). Post-hoc Sidak's multiple comparisons tests showed a significant increase in firing frequency with application of uPSEM$^{792}$ (compared with aCSF) in PSAM$^4$-GlyR$^+$ but not control neurons. In contrast, high potassium after uPSEM$^{792}$ caused a significant increase in firing frequency in control neurons (compared to aCSF and uPSEM$^{792}$), as opposed to reduced firing frequency in PSAM$^4$-GlyR$^+$ neurons, likely because of depolarization block, as is also suggested by the decrement in the amplitude of action currents with uPSEM$^{792}$-induced firing (*Figure 4C*).

Taken together, the whole-cell and cell-attached recording results demonstrate that (1) activation of PSAM$^4$-GlyR significantly depolarized the membrane potential, (2) the magnitude of the decrease in membrane resistance produced by opening of PSAM$^4$-GlyR channels was not sufficient to inhibit neuronal firing via electrical shunting, and (3) PSAM$^4$-GlyR activation could trigger action potential firing when neurons are depolarized to subthreshold potentials.

## Sustained activation of PSAM$^4$-GlyR results in runaway depolarization of D1-MSNs

Depolarization from the resting membrane potential ($-83$ mV) by activation of PSAM$^4$-GlyR was expected given the calculated reversal potential of the channel ($-62$ mV, see Materials and methods). However, in most D1-MSNs ($\sim$57%), the depolarization overshot the expected reversal potential to potentials more positive than $-62$ mV (*Figure 3A*). Therefore, we

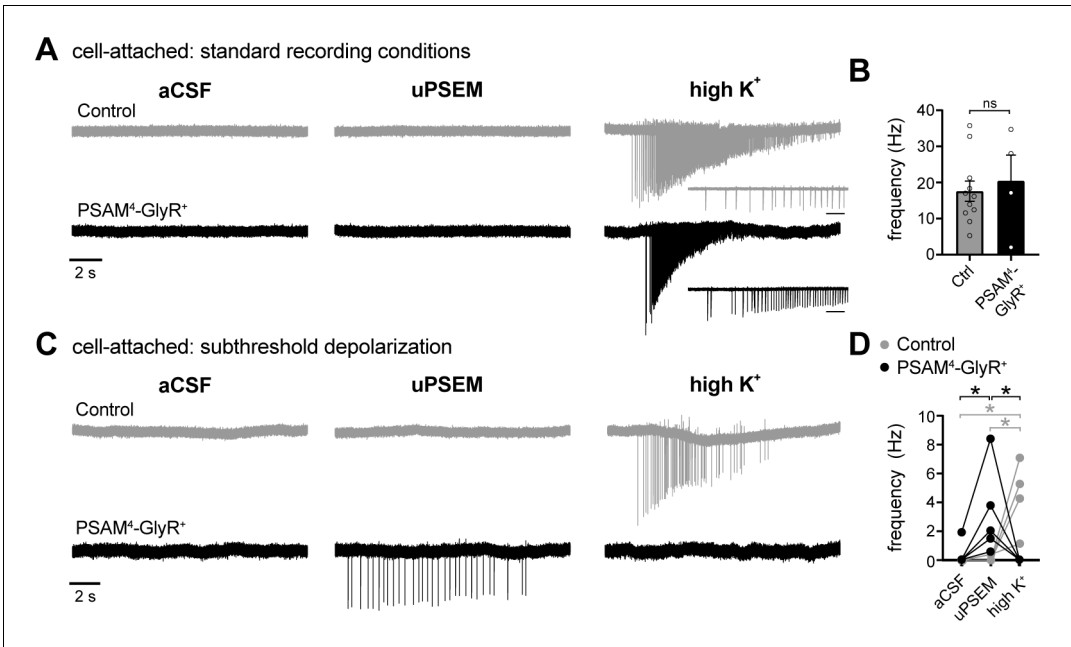

**Figure 4.** In cell-attached configuration, activation of PSAM⁴-GlyR triggers firing of D1-MSNs when the membrane is depolarized to subthreshold potentials. (**A**) Representative traces of cell-attached recordings from a control (gray) or PSAM⁴-GlyR⁺ neuron in standard aCSF (4.5 mM [K⁺]). Action currents were evoked following uPSEM⁷⁹² by applying a high external concentration of potassium (high K⁺, ~40 mM). The insets below the high K⁺ traces represent an expanded timescale at the beginning of firing; Inset scale bar = 200 ms. (**B**) In PSAM⁴-GlyR⁺ neurons, activation of PSAM⁴-GlyR with uPSEM⁷⁹² had no effect on the frequency of action currents in response to high K⁺. (**C**) Representative traces of cell-attached recordings from a control (gray) or PSAM⁴-GlyR⁺ neuron in aCSF containing 10–13 mM [K⁺]. In PSAM⁴-GlyR⁺ neurons, activation of PSAM⁴-GlyR with uPSEM⁷⁹² produced firing. There was no further firing with application of high K⁺. uPSEM⁷⁹² had no effect on firing in control neurons. (**D**) Plot of the frequency of action currents in control and PSAM⁴-GlyR⁺ neurons in aCSF, uPSEM⁷⁹² and high K⁺ (mixed effect two-way ANOVA with Sidak's multiple comparisons test). Line and error bars represent means ± SEM. * indicates statistical significance, ns denotes not significant.

The online version of this article includes the following source data for figure 4:

**Source data 1.** PSAM4-GlyR induces D1-MSN firing in cell-attached recordings (source data).

---

examined the current–voltage relationship of PSAM⁴-GlyR. Voltage steps (from $V_{hold}$ −88 mV, 1 s, −118 to −28 mV, 10 mV increments) were made in aCSF and uPSEM⁷⁹², and PSAM⁴-GlyR current was isolated by subtraction (**Figure 5A**). In control neurons, uPSEM⁷⁹² did not produce substantial current across the voltage range (**Figure 5A and C**). In PSAM⁴-GlyR⁺ neurons, uPSEM⁷⁹² produced a transient current with the onset of the voltage step (inset '1', **Figure 5A**) that relaxed toward a steady-state current during the voltage step (inset '2', **Figure 5A**). The transient current reversed polarity at −61.4 mV (*n* = 21, CI: −64.0 to −59.0 mV, linear regression, $r^2$ = 0.58, $F_{1, 103}$ = 142.5, p<0.0001), consistent with the calculated reversal potential of the PSAM⁴-GlyR channel. Transient PSAM⁴-GlyR current was inward at potentials more negative than −61 mV, and outward at potentials less negative than −61 mV (**Figure 5C**). At steady state, the reversal potential of PSAM⁴-GlyR current showed a significant rightward shift to −44.2 mV (*n* = 21, CI: −48.3 to −38.2 mV, linear regression, $r^2$ = 0.39, $F_{1, 103}$ = 66.9, p<0.0001; paired *t*-test, *t* = 5.73, p<0.0001, **Figure 5D and E**). Further, the steady-state current showed a U-shaped current–voltage relationship (**Figure 5D**), and the outward current component was significantly smaller than the transient outward current (49.8 ± 38.4 pA *vs.* 307.1 ± 62.0 pA at $V_{hold}$ −28 mV respectively, unpaired *t*-test, p<0.0001; **Figure 5D**). Thus, in D1-MSNs, the majority of outward current carried by PSAM⁴-GlyR channels was transient. Hence, during the firing cycle, membrane hyperpolarization via PSAM⁴-GlyR activation is expected to be minimal or transient, and thus, PSAM⁴-GlyR activation is predominantly depolarizing.

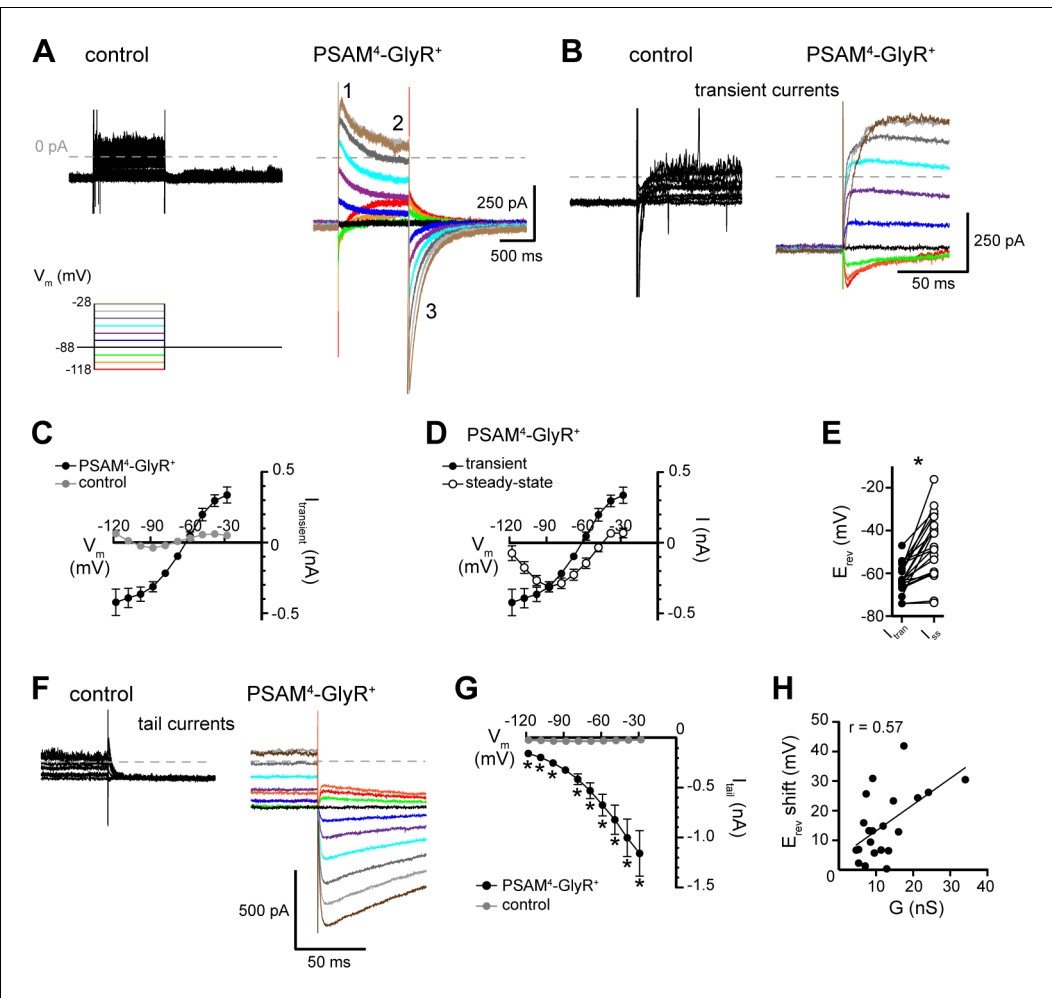

**Figure 5.** Current–voltage relationship of PSAM[4]-GlyR current shows rightward shift in reversal potential and transient or small outward currents at depolarized potentials. (**A**) Current was recorded in response to voltage steps (1 s, −118 to −28 mV, 10 mV) from $V_{hold}$ −88 mV in aCSF and after application of uPSEM[792] (50 or 100 nM). uPSEM[792]-induced current was isolated by subtraction. Representative traces of uPSEM[792]-induced current during voltage steps in a control (left) and a PSAM[4]-GlyR[+] neuron (right). Inset numbers by trace indicate transient (**Roth, 2016**), steady-state (**Saloman et al., 2016**), and tail (**Gomez et al., 2017**) currents. (**B**) Expanded timescale of transient currents (inset '1' in **A**) in a control and a PSAM[4]-GlyR[+] neuron. (**C**) Plot of current–voltage relationship of the transient current ($I_{transient}$) in control ($n$ = 11) *vs.* PSAM[4]-GlyR[+] neurons ($n$ = 21). The reversal potential of transient current in PSAM[4]-GlyR[+] neurons was −61 mV. (**D**) Plot of current–voltage relationship of the transient (inset '1' in **A**, same as in **C**) *vs.* steady-state uPSEM[792]-induced (inset '2' in **A**) currents in PSAM[4]-GlyR[+] neurons. At steady state, the reversal potential of uPSEM[792]-induced current shifted to −44 mV. (**E**) Plot of the pairwise shift of reversal potential ($E_{rev}$) between transient ($I_{tran}$) and steady-state currents ($I_{ss}$). (**F**) Expanded timescale of tail currents (inset '3' in **A**) in a control and a PSAM[4]-GlyR[+] neuron. (**G**) Plot of the amplitude of tail currents measured at $V_{hold}$ −88 mV ($I_{tail}$) versus holding potential of the preceding voltage step in control and PSAM[4]-GlyR[+] neurons. uPSEM[792]-induced tail current at −88 mV demonstrated robust augmentation with prior depolarization and reduction with prior hyperpolarization. (**H**) The magnitude of the rightward shift in reversal potential between transient and steady-state current was positively correlated with the PSAM[4]-GlyR conductance (G in nS). Line and error bars represent means ± SEM, * indicates statistical significance.

The online version of this article includes the following source data for figure 5:

**Source data 1.** Current-voltage relationship of PSAM4-GlyR current (source data).

## Instability of chloride equilibrium underlies PSAM$^4$-GlyR-mediated depolarization

In order to gain a mechanistic understanding of PSAM$^4$-GlyR-mediated depolarization, we examined tail currents generated by stepping back to a holding potential of −88 mV from preceding voltage steps. In principle, relaxation of outward currents during voltage steps could reflect fewer available channels (e.g., channel desensitization, inactivation, or voltage-dependent block), which would be expected to result in smaller tail currents. Alternatively, the relaxation of outward currents could be a manifestation of reduced driving force of chloride influx through PSAM$^4$-GlyR channels due to the accumulation of intracellular chloride over the course of the voltage steps, which would be expected to result in larger tail currents. Indeed, many reports show that sustained activation of chloride conductances including GABA$_A$ receptors (*Huguenard and Alger, 1986*; *Staley et al., 1995*; *Thompson and Gähwiler, 1989*), glycine receptors (*Karlsson et al., 2011*), and Halorhodopsin (*Raimondo et al., 2012*; *Alfonsa et al., 2015*) can lead to chloride influx that overwhelms homeostatic mechanisms to pump chloride out, thereby elevating the intracellular chloride concentration. An increase in intracellular chloride manifests as a rightward shift of the chloride reversal potential, commensurate with our observation of a rightward shift of the PSAM$^4$-GlyR reversal potential from −61 mV to −44 mV. Further, the magnitude of the tail current increased substantially with preceding depolarizing steps (*Figure 5F and G*; RM two-way ANOVA, significant PSAM$^4$-GlyR activation effect: $F_{1, 30}$ = 18.05, p=0.0002; significant PSAM$^4$-GlyR activation × voltage step interaction: $F_{9, 270}$ = 9.55, p<0.0001; Dunnett's multiple comparisons test shows that tail currents from every preceding voltage step were significantly different from current at −88 mV without a preceding voltage step). These data reveal that the decline in outward current was not due to fewer available channels but was consistent with a change in the driving force and shift in the reversal potential during the depolarizing voltage steps. In addition, the magnitude of the shift in reversal potential was positively correlated with the conductance of PSAM$^4$-GlyR, measured at $V_{hold}$ −88 mV (Pearson's correlation, r = 0.57, p=0.0008, *n* = 21; *Figure 5H*). The tail current analysis also suggested some intrinsic voltage-sensitivity of PSAM$^4$-GlyR. The magnitude of tail currents decreased with preceding hyperpolarizing steps (e.g., −98, −108, or −118 mV) and increased with preceding depolarizing steps (e.g., −78 or −68 mV; *Figure 5F and G*), suggesting that hyperpolarization reduced the number of available channels, or that depolarization increased the number of available channels. Taken together, the results suggest that activation of PSAM$^4$-GlyR in D1-MSNs is largely depolarizing at sub- and peri-threshold membrane potentials likely due to two mechanisms: increased channel availability with depolarization due to intrinsic voltage-sensitive mechanisms, and more prominently, a rightward shift of PSAM$^4$-GlyR reversal potential due to accumulation of intracellular chloride at depolarized potentials.

## Loss of GABAergic inhibition following activation of PSAM$^4$-GlyR

Shifts in intracellular chloride concentration have been shown to result in 'apparent cross-desensitization' of glycine- and GABA$_A$ receptors whereby sustained activation of one chloride-permeable channel (e.g., glycine channel) results in reduced currents carried by another chloride-permeable channel (e.g., GABA$_A$ channel) and vice versa (*Karlsson et al., 2011*). Therefore, we examined GABA$_A$ receptor-mediated synaptic currents in PSAM$^4$-GlyR$^+$ neurons following activation of PSAM$^4$-GlyR. A bipolar stimulating electrode was placed in the brain slice and GABA$_A$ receptor-mediated synaptic currents were electrically evoked. The same voltage steps were made (as to assess the current–voltage relationship of PSAM$^4$-GlyR current) and GABA$_A$ receptor-mediated synaptic currents were evoked 850 ms into the voltage step where uPSEM$^{792}$ current reached steady state (single electrical pulse, 0.1 ms, *Figure 6A and B*). In neurons that showed a rightward shift in reversal potential of PSAM$^4$-GlyR (>5 mV shift, *n* = 5, $E_{rev}$shift = 17.3 ± 6.3 mV), the amplitude of the GABA$_A$ receptor-mediated synaptic currents significantly decreased upon activation of PSAM$^4$-GlyR compared with aCSF (−33.4 ± 6.0 pA *vs.* −7.8 ± 1.3 pA at $V_{hold}$ −88 mV, paired *t*-test: p=0.01; *Figure 6C*). Further, the reversal potential of GABA$_A$ receptor-mediated synaptic currents showed a significant rightward shift from −60.9 mV in aCSF to −51.6 mV upon activation of PSAM$^4$-GlyR (aCSF: CI: −64.2 to −58.1 mV, linear regression, $r^2$ = 0.78, $F_{1, 13}$ = 43.7, p<0.0001; uPSEM$^{792}$: CI: −55.2 to −46.7 mV, $r^2$ = 0.66, $F_{1, 7}$ = 13.7, p=0.008; test of equal intercepts: $F_{1, 21}$ = 18.3, p=0.0003; *Figure 6C*). In neurons that did not show a shift in reversal potential of PSAM$^4$-GlyR

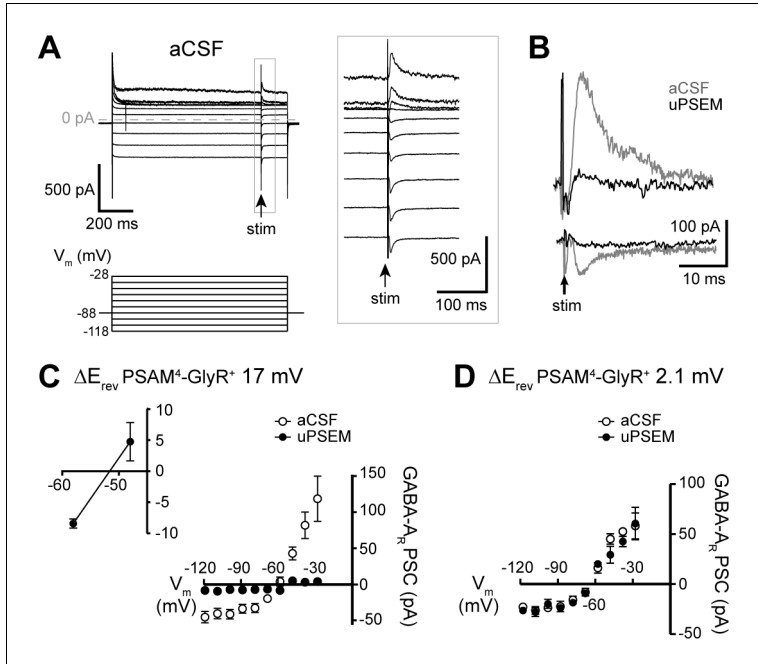

**Figure 6.** PSAM[4]-GlyR activation in D1-MSNs reduces GABAergic synaptic inhibition. (**A**) Representative traces of GABA-A receptor-mediated synaptic currents in a PSAM[4]-GlyR[+] neuron in aCSF, evoked during voltage steps from $V_{hold}$ −88 mV (voltage steps: 1 s, −118 to −28 mV, 10 mV; electrical stimulation at 850 ms into voltage steps). Panel on the right shows the traces in expanded timescale corresponding to the gray rectangle on the left. (**B**) Representative traces of GABA-A receptor-mediated currents evoked in the same neuron from $V_{hold}$ −28 mV (outward) and −118 mV (inward) in aCSF (gray) and uPSEM[792] (black). (**C**) Plot of the current–voltage relationship of the GABA-A receptor synaptic currents in aCSF and uPSEM[792] in neurons where the reversal potential of PSAM[4]-GlyR current shifted by >5 mV (*n* = 5, average shift of 17.0 mV). The reversal potential of GABA-A receptor-mediated current was −60.9 mV in aCSF and −51.6 mV in uPSEM[792]. Inset (top left) shows values of GABA-A receptor synaptic currents in uPSEM[792] near the reversal potential in an expanded scale. (**D**) Plot of the current–voltage relationship of the GABA-A receptor-mediated synaptic current in aCSF and uPSEM[792] in neurons where the reversal potential of PSAM[4]-GlyR current did not shift (*n* = 3, average shift of 2.1 mV). The reversal potential of GABA-A receptor-mediated synaptic currents was similar in aCSF (−64.7 mV) and uPSEM[792] (−64.9 mV). Line and error bars represent means ± SEM, * indicates statistical significance.

The online version of this article includes the following source data for figure 6:

**Source data 1.** Reduced GABAergic synaptic inhibition (source data).

(*n* = 3, $E_{rev}$ shift = 2.1 ± 1.0 mV), there was no change in the reversal potential or amplitude of $GABA_A$ receptor-mediated synaptic currents (aCSF: $GABA_A E_{rev}$ = −64.7 mV, CI: −68.5 to −62.1 mV, linear regression, $r^2$ = 0.91, $F_{1, 7}$ = 68.8, p<0.0001; uPSEM[792]: $GABA_A E_{rev}$ = −64.9 mV, CI: −66.7 to −63.4 mV, $r^2$ = 0.96, $F_{1, 4}$ = 87.5, p=0.0007; *Figure 6D*). These results indicate that activation of PSAM[4]-GlyR reduced GABAergic inhibition of D1-MSNs.

## Discussion

Strategies to manipulate neuronal activity in defined cell populations have become instrumental in mapping neural circuits and correlating neuronal and circuit activity with behavior. Engineered receptors to promote excitation (e.g., Channelrhodopsin-2, $G_q$-DREADDs) have been largely successful, albeit with careful consideration of ligand-related off-target effects (*Gomez et al., 2017*; *Manvich et al., 2018*). In contrast, silencing of neuronal firing has been uniquely challenging, especially when relying on chloride conductances (*Wiegert et al., 2017*). The utility of a chloride conductance like PSAM[4]-GlyR to silence firing relies on hyperpolarization of the membrane at potentials more positive than the chloride reversal potential and/or the efficacy of the channels to shunt membrane depolarization (*Doyon et al., 2016*).

The present study shows that PSAM[4]-GlyR in D1-MSNs is predominantly depolarizing, fails to inhibit neuronal firing, triggers action potential firing when neurons are depolarized to subthreshold potentials, and has a limited capacity to hyperpolarize these neurons even at more depolarized potentials. When activated near the resting membrane potential (−83 mV), PSAM[4]-GlyR passes inward current and depolarizes D1-MSNs. When activated at potentials more depolarized than the chloride reversal potential, PSAM[4]-GlyR passes transient or small outward currents, likely due to influx and accumulation of intracellular chloride that overwhelms endogenous mechanisms to restore the chloride gradient (*Huguenard and Alger, 1986*; *Staley et al., 1995*; *Thompson and Gähwiler, 1989*; *Karlsson et al., 2011*; *Raimondo et al., 2012*; *Alfonsa et al., 2015*). Bath application or systemic injection of high potency exogenous agonists of chloride conductances (e.g., uPSEM[792]) is a very different phenomenon than physiological activation of native chloride conductances (e.g., $GABA_A$ and glycine receptors) by endogenous agonists. The release and clearance of endogenous neurotransmitters are usually tightly controlled and occur over the scale of milliseconds, allowing the neurons to better maintain a stable intra- and extracellular chloride gradient. This is unlike exogenous agonists that result in sustained activation over seconds or minutes and cause massive chloride influx and a shift in chloride reversal potential. In addition, the expression of endogenous chloride conductances is also regulated, unlike viral overexpression of exogenous conductances (e.g., PSAM[4]-GlyR) which only augments the magnitude of chloride influx, hence the unpredictable nature of neuronal responses to the activation of exogenous chloride conductances with exogenous agonists.

More commonly, chloride conductances inhibit action potential firing through electrical shunting. Opening channels decrease membrane resistance which reduces (shunts) depolarization in response to inward current. While there was a significant decrease in membrane resistance associated with opening PSAM[4]-GlyR in D1-MSNs (~30% reduction), it was not sufficient to prevent suprathreshold depolarization by current injection. The majority of neurons were equally capable of firing action potentials to current injection. Despite using high viral titer in our experiments, the decrease in membrane resistance we observed was lower than previously reported in cortical layer 2/3 neurons where PSAM[4]-GlyR activation suppressed firing (*Magnus et al., 2019*). While increasing PSAM[4]-GlyR conductance by further increasing expression levels could theoretically improve shunting efficacy, the data in this study show a strong correlation between PSAM[4]-GlyR conductance and the rightward shift in PSAM[4]-GlyR reversal potential, suggesting that increasing expression levels will only promote further depolarization with PSAM[4]-GlyR activation. The lack of shunting efficacy in D1-MSNs is likely due to their low membrane resistance (~78 MΩ in brain slices as measured above) relative to other neurons in the brain (e.g., pyramidal neurons in cortex and hippocampus). It is important to note that membrane resistance is inversely proportional to temperature, age, and number/activity of synaptic inputs (*Kroon et al., 2019*; *Waters and Helmchen, 2006*; *Fernandez et al., 2018*), and therefore, the shunting efficacy of PSAM[4]-GlyRs during behavioral experiments in vivo is likely to be lower than in brain slices.

In the awake behaving state in vivo, MSNs show brief and irregular spontaneous membrane potential fluctuations between depolarizations (−50 to −60 mV) that facilitate action potential firing and hyperpolarizations (−80 to −90 mV), with the overall membrane potential following a Gaussian distribution (mean ~ −69 mV) (*Mahon et al., 2006*). The cell-attached data show that activation of PSAM[4]-GlyR triggers action potential firing when the neurons are depolarized to subthreshold potentials by increasing extracellular potassium. A caveat of these data is that increasing extracellular potassium may indirectly impair chloride extrusion via the potassium-chloride cotransporter, KCC2, which may exacerbate instability of the chloride gradient and make the cells more excitable. Such KCC2 perturbance is expected to affect both control and PSAM[4]-GlyR cells. However, the increased excitability observed in our experiments was limited to the PSAM[4]-GlyR cells supporting the conclusion of PSAM[4]-GlyR driven increase in excitability. The current-clamp data showed that activation of PSAM[4]-GlyR depolarized D1-MSNs by ~20 mV and was sufficient to produce depolarization block in a proportion of neurons. This implies that in vivo, PSAM[4]-GlyR activation will effectively increase the probability and duration of spontaneous depolarizations, and increase the probability of D1-MSN firing especially during these depolarizations. The results also show that PSAM[4]-GlyR activation results in 'apparent cross-desensitization' of $GABA_A$ synaptic currents, producing a rightward shift in the $GABA_A$ reversal potential and a profound reduction of inhibitory $GABA_A$ synaptic currents onto D1-MSNs, likely attributed to electrical shunting since the reduction

was observed at all potentials. Given the critical role of local GABAergic inhibition in the striatum in regulating neuronal activity (*Burke et al., 2017*; *Koós and Tepper, 1999*), PSAM$^4$-GlyR triggered reduction of GABAergic inhibition will further increase the likelihood of D1-MSNs activation rather than inhibition. Thus, increased c-fos expression in D1-MSNs following in vivo activation of PSAM$^4$-GlyR is likely caused by PSAM$^4$-GlyR-induced depolarization and firing in addition to loss of GABAergic inhibition, and reflects increased neuronal activity of D1-MSNs in vivo.

Finally, PSAM$^4$-GlyR-induced depolarization may allow robust calcium influx via voltage-gated calcium channels and NMDA receptors relieved from voltage-dependent magnesium pore-block (*Mayer and Westbrook, 1987*; *Mayer et al., 1984*). Calcium influx activates calcium/calmodulin-dependent protein kinases (CaMKs), calcium response elements in genes, and various other signaling cascades (*Pasek et al., 2015*; *Clapham, 2007*), which are known to influence synaptic plasticity and behavior (*Lisman et al., 2002*; *Wayman et al., 2008*). Therefore, it is possible that when used to silence neurons, PSAM$^4$-GlyR activation could confound the interpretation of experimental results due to depolarization-induced calcium influx, independent of PSAM$^4$-GlyR's effect on action potential firing like in the case of depolarization-block.

In summary, activation of PSAM$^4$-GlyR expressed in D1-MSNs in the ventral striatum enhanced neuronal activity through direct depolarization and did not suppress action potential firing via membrane shunting. The results of our study show that the PSAM$^4$-GlyR approach to silence neurons may not be suitable for all cell types and highlight the need to validate the inhibition of neuronal firing by PSAM$^4$-GlyR in the cell type of interest prior to behavioral studies. More broadly, these data demonstrate that achieving neuronal silencing with chloride conductances continues to be challenging and may result in unexpected neuronal activation.

## Materials and methods

### Subjects

All experimental procedures were conducted in accordance with the guidelines of the National Institutes of Health Guide for the Care and use of Laboratory Animals, and approved by the Animal Care and Use Committee of the National Institute on Drug Abuse. We used adult female and male prodynorphin-Cre mice (pdyn-Cre; 8–12 weeks old, breeding facility at the Intramural Research Program, National Institute on Drug Abuse) for electrophysiology and c-fos experiments. We crossed pdyn-Cre and Ai9 Rosa-tdTomato mice to validate selectivity of viral transduction (*Figure 1A*). Mice were group housed four per cage and maintained under a 12 hr light cycle at $21 \pm 2°C$. Food and water were freely available.

### Stereotaxic virus injection

AAV-SYN-flex-PSAM$^4$-GlyR-IRES-EGFP was a gift from Scott Sternson (Addgene viral prep # 119741-AAV5; http://n2t.net/addgene:119741; RRID:Addgene_119741; >1 × 10$^{13}$ vg/ml). Control virus was AAV1-EF1α-DIO-eYFP (>1 × 10$^{13}$ vg/ml, University of North Carolina Viral Core, NC). Mice were anesthetized with an i.p. injection of a cocktail of ketamine (100 mg/kg) and xylazine (10 mg/kg) and secured in a stereotaxic frame (David Kopf Instruments, CA). PSAM$^4$-GlyR and control viruses were injected (0.2–0.5 µl, 0.05 µl/min) into the medial ventral striatum (targeted coordinates relative to bregma: 1.4 mm AP, 0.5 mm ML (10°), and −4.5 mm DV) using a 29 G stainless steel cannula connected to a 2 µl Hamilton syringe or a Nanoject system (Drummond Scientific, PA). The injectors were retracted slowly after 5 min. Mice were given carprofen (5 mg/kg) post-surgery for pain relief.

### Brain slice preparation and electrophysiological recordings

After >4 weeks of viral incubation, mice were anesthetized with Euthasol (i.p., Virbac AH, Inc, TX) and then decapitated. Brains were rapidly removed and placed in room temperature (25°C) modified Krebs' buffer containing (in mM): 125 NaCl, 4.5 KCl, 1.0 MgCl$_2$, 1.2 CaCl$_2$, 1.25 NaH$_2$PO$_4$, 11 D-glucose, and 23.8 NaHCO$_3$, bubbled with 95/5% O$_2$/CO$_2$ with 5 µM MK-801 to increase slice viability. Using a vibrating microtome (Leica Biosystems, IL). Coronal brain slices (220 µm) containing the ventral striatum were collected and incubated at 32°C for 4 min, then transferred to room temperature until use. All recordings were from neurons in the ventro-medial shell of the nucleus accumbens.

Cells were visualized using IR-DIC and fluorescence on an upright Olympus BX51WI microscope (Olympus, MA). Transduced D1-MSNs were identified by visualization of eGFP or eYFP. Whole-cell patch-clamp recordings were made at 35 ± 1°C with a Multiclamp 700B amplifier and Digidata 1440a digitizer with Clampex and Axoscope software (Molecular Devices, CA). Data were digitized at 20 kHz. For voltage-clamp experiments, data were filtered at 2 kHz. Recordings were performed in modified Krebs' buffer. In the majority of experiments, receptor antagonists were used to eliminate fast synaptic transmission (3 μM NBQX or DNQX and 100 μM picrotoxin). For whole-cell configuration recordings, pipette resistance was 1.5–2.5 MΩ when filled with internal solution containing (in mM) K-methylsulfate (122), HEPES (*Chung, 2015*), EGTA (0.45), NaCl (*Krashes et al., 2014*), MgCl$_2$ (1.8), CaCl$_2$ (0.1), Mg-ATP (*Manvich et al., 2018*), Na-GTP (0.3), creatine phosphate disodium (*Enoksson et al., 2012*), pH 7.35 and 284 mOsm. For cell-attached configuration recordings, pipette resistance was 2–4 MΩ when filled with Krebs' buffer. Cell-attached recordings in which the neuron did not fire under uPSEM[792] or high [K$^+$] were excluded from analysis.

Assuming permeability to Cl$^-$ only, the calculated reversal potential of PSAM[4]-GlyR was −62.3 mV. Assuming that PSAM[4]-GlyR has similar Cl$^-$/HCO$_3^-$ permeability to GlyR (P$_{HCO3-}$:P$_{Cl-}$=0.14) (*Jun et al., 2016*), the calculated reversal potential of PSAM[4]-GlyR in our recording conditions was −60.7 mV ([Cl$^-$]$_{out}$133.9 mM, [Cl$^-$]$_{in}$12.8 mM, [HCO$_3^-$]$_{out}$23.8 mM, and assuming [HCO$_3^-$]$_{in}$8 mM). Series resistance was monitored throughout the recordings and not compensated. Reported voltages were corrected for a liquid junction potential of −8 mV between the internal and external solutions. To measure neuronal excitability, current was injected in 50 pA increments (2 s). In determining the current–voltage (I-V) relationship using voltage steps, the external solution of some recordings included 1 μM tetrodotoxin (TTX) to eliminate sodium channel-dependent spiking. There were no differences in the shape of the I-V with or without TTX, so data were combined. Reversal potentials were determined by linear regression considering each replicate an individual point. Recordings in which current did not cross 0 pA were omitted from analysis ($n$ = 2). GABA$_A$ receptor-mediated synaptic currents were evoked with electrical stimulation delivered by a bipolar stimulating electrode placed in the brain slice, in the presence of ionotropic glutamate receptor antagonists (NBQX/DNQX) to isolate fast GABAergic synaptic transmission. Peak amplitude of GABA$_A$ receptor-mediated synaptic currents was measured 3–4 ms from the apparent peak to remove any potential contribution from a stimulation-induced 'escaped' action potential.

## Immunohistochemistry and confocal microscopy

For immunohistochemistry, confocal microscopy, and image analysis, the experimenter was blinded to the animal treatment group. Mice microinjected with PSAM[4]-GlyR *vs.* control AAV into the medial ventral striatum received an i.p. injection of uPSEM[792] (3 mg/kg) or saline followed 30 min later by an i.p. injection of fentanyl (0.2 mg/kg) or saline (*Figure 1B*). After 90 min, at the expected peak of c-fos protein expression (*Barros et al., 2015*), mice were euthanized with Euthasol (i.p.) and transcardially perfused with 1× PBS followed by ice-cold 4% paraformaldehyde (pH 7.4, Sigma Aldrich, MO). The brains were collected and fixed overnight in 4% paraformaldehyde at 4°C, and then moved to PBS. Coronal brains slices (50 μm) containing the viral injection sites were collected using a vibratome (Leica Biosystems, IL). Free-floating brain slices were washed in PBS (×3), permeabilized and blocked in a solution containing PBS, 0.3% Triton-X, and 5% normal donkey serum for 2 hr. Slices were then incubated at 4° overnight in a solution containing PBS, 0.03% Triton-X, 5% normal donkey serum, and 1:4000 rabbit anti-c-fos primary antibody (Cell Signaling, MA). This was followed by a wash in PBS (×3) and incubation in 1:500 donkey anti-rabbit secondary antibody, conjugated to Alexa-Fluor-647 (Jackson ImmunoResearch, PA) for 1 hr and 45 min. Slices were then washed in PBS (×3) and mounted with DAPI Fluoromount mounting medium (Thermo Fisher Scientific, MA). Confocal images were collected using an Olympus Fluoview FV1000 confocal microscope with 20× (0.75 NA) or 40× (0.95 NA) objective lens and processed using ImageJ. Z-stack images of the viral injection site in the right hemisphere (two to three brain sections) were collected. Expression of c-fos and eGFP/eYFP was manually quantified by counting fluorescent cells within the image frame. The number of non-transduced cells was obtained by subtracting the number of transduced cells from the number of cells stained with DAPI. Results from all brain sections were averaged for each mouse.

Drugs uPSEM[792], MK-801, picrotoxin, and NBQX were purchased from Tocris (MN). All other electrophysiological reagents were obtained from Sigma-Aldrich (MO).

## Data analysis

Data were analyzed using Clampfit 10.7. Data are presented as representative traces, or in bar graphs with means ± SEM, and in scatter plots where each point is an individual mouse (*Figure 1*) or cell (*Figures 2–6*). In *Figure 1*, n = number of mice (8–12/group). In *Figures 2–6*, n = number of cells. The number of mice used is as follows: *Figure 2* (5–8/group), *Figure 3* (3–7/group), *Figure 4* (2–4/group), *Figure 5* (4–7/group), and *Figure 6* (*Manvich et al., 2018*). No sample was tested in the same experiment more than once (technical replication). Image analysis for *Figure 1* was done blindly. Data were analyzed using GraphPad Prism (v 8.1.1; GraphPad software, CA). The statistical tests we used included *t*-tests (paired or unpaired), ANOVAs (with or without repeated measures) with Sidak's or Dunnett's multiple comparison test (as recommended by Prism), ANOVA mixed-effects analysis (when missing values prohibited repeated-measures analysis with ANOVAs), Pearson's correlation, and linear regression (we ensured there were no departures from linearity with replicates test). Significance level was set at $p < 0.05$.

## Acknowledgements

We would like to thank Dr. Shiliang Zhang and the Confocal and Electron Microscopy Core at the NIDA Intramural Research Program for helping with confocal imaging. This project was supported by the Intramural Research Program at the National Institute on Drug Abuse, DA048085 (KM), and the Center on Compulsive Behaviors, National Institutes of Health via NIH Director's Challenge Award (SCG).

## Additional information

### Funding

| Funder | Grant reference number | Author |
|---|---|---|
| National Institute on Drug Abuse | DA048085 | Khaled Moussawi |

The funders had no role in study design, data collection and interpretation, or the decision to submit the work for publication.

### Author contributions

Stephanie C Gantz, Conceptualization, Data curation, Formal analysis, Visualization, Writing - original draft, Writing - review and editing; Maria M Ortiz, Formal analysis, Investigation, Writing - original draft; Andrew J Belilos, Formal analysis, Investigation; Khaled Moussawi, Conceptualization, Resources, Data curation, Software, Formal analysis, Supervision, Funding acquisition, Validation, Investigation, Visualization, Methodology, Writing - original draft, Project administration, Writing - review and editing

### Author ORCIDs

Stephanie C Gantz https://orcid.org/0000-0002-1800-4400
Khaled Moussawi https://orcid.org/0000-0001-6378-0428

### Ethics

Animal experimentation: All experimental procedures were conducted in accordance with the guidelines of the National Institutes of Health Guide for the Care and use of Laboratory Animals, and approved by the Animal Care and Use Committee of the National Institute on Drug Abuse (Protocol #17-CNRB-133 ).

### Decision letter and Author response

Decision letter https://doi.org/10.7554/eLife.64241.sa1
Author response https://doi.org/10.7554/eLife.64241.sa2

## Additional files

### Supplementary files
• Transparent reporting form

### Data availability
All data generated or analyzed during this study are included in the manuscript and supporting files.

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
