## [Decision Letter]

**Acceptance summary:**

Ganz et al. demonstrate that the ultrapotent chemogenic silencing strategy utilising PSAM^4^-GlyR can excite D1-expressing medium spiny neurons. This highlights the need to validate the inhibition of neuronal firing by PSAM^4^-GlyR in the brain area and cell-type of interest. This work shows that neuronal silencing with chloride conductances can in some cases result in unexpected neuronal activation. This is likely of interest to those designing experiments which utilise the latest chemogenic silencing strategies.

**Decision letter after peer review:**

Thank you for submitting your article "Excitation of medium spiny neurons by 'inhibitory' ultrapotent chemogenetics via shifts in chloride reversal potential" for consideration by *eLife*. Your article has been reviewed by 3 peer reviewers, including Joseph V Raimondo as the Reviewing Editor and Reviewer #1, and the evaluation has been overseen by John Huguenard as the Senior Editor. The following individuals involved in review of your submission have agreed to reveal their identity: Ofer Yizhar (Reviewer #2); Melanie A Woodin (Reviewer #3).

The reviewers have discussed the reviews with one another and the Reviewing Editor has drafted this decision to help you prepare a revised submission.

This manuscript performs a much-needed validation of a new generation of chemogenetic tools based on the PSAM family of ligand-gated chloride channels. The authors expressed PSAM^4^-GlyR in D1 MSNs in striatum. They then perform c-Fos staining after systemic injection of the ligand (uPSEM^792^) and show that paradoxically, c-Fos expression is elevated in the expressing cells. This is followed bys a detailed and thorough investigation of the electrophysiological effects of uPSEM^792^ delivery in acute brain slices. The authors find that uPSEM mostly depolarizes the PSAM-expressing neurons, which is expected from the chloride reversal potential in these cells, but also fails to shunt action potential firing. They further demonstrate that prolonged activation of the PSAM channels leads chloride accumulation and a further shift of the chloride reversal and a loss of GABAa-mediated inhibition in the same cells. Altogether, their findings provide a mechanistic explanation for the paradoxical excitatory effects of PSAM^4^-GlyR in D1 MSNs in-vivo. This amounts to a cautionary tale regarding the need to validate the effects of chemogenetic or optogenetic tools prior to their application in behavioral experiments.

The reviewers deemed your work of sufficient interest. We felt that the key finding was the raised c-Fos expression observed in D1 MSNs following activation of PSAM^4^-GlyRs in vivo. However, multiple reviewers raised concerns about the N and the number of mice used not being entirely clear. In particular in Figure 1D the experimental groups have only 5 data points (presumably 5 mice). Considering that the paper is resting on this important finding, the reviewing team asks that you focus your revision on increasing the N for the c-fos experiment to 8-10 animals for the experimental groups to confirm the validity of this key finding.

Reviewer #1:

The study is well performed.

Reviewer #2:

This manuscript performs a much-needed validation of a new generation of chemogenetic tools developed by the Sternson lab, based on the PSAM family of ligand-gated chloride channels that was first described by the same group in 2011. The authors expressed PSAM^4^-GlyR in D1 MSNs in striatum. They then perform c-Fos staining after systemic injection of the ligand (uPSEM^792^) and show that paradoxically, c-Fos expression is elevated in the expressing cells. What follows is a detailed and thorough investigation of the electrophysiological effects of uPSEM^792^ delivery in acute brain slices. The authors find that uPSEM mostly depolarizes the PSAM-expressing neurons, which is expected from the chloride reversal potential in these cells, but also fails to shunt action potential firing. They further demonstrate that prolonged activation of the PSAM channels leads to a further shift of the chloride reversal potential to more depolarized potentials and a loss of GABAa-mediated inhibition in the same cells. Altogether, their findings provide a mechanistic explanation for the surprising in-vivo finding demonstrated in Figure 1 and amount to a cautionary tale regarding the need to validate the effects of chemogenetic or optogenetic tools prior to their application in behavioral experiments. The study is well-done and written in a clear and concise manner. It would be of importance to many scientists working in this field, particularly since few studies have been published utilizing these new potentially powerful tools for chemogenetic inhibition. I support the publication of this work in *eLife*.

Reviewer #3:

Since the initial report of DREADDs (Armbruster 2007, PNAS), chemogenetics has received significant attention for their ability to remotely control neuronal activity. This has included the development of additional chemogenetic tools (proteins and agonists), such as the PSAM/PSEM. The main objective of this study is to validate the use of PSAM^4^-GlyR activated by uPSEM^792^ as an inhibitory chemogenetic tool. By performing extensive electrophysiological characterization, the authors conclude that PSAM^4^-GlyR activated by uPSEM^792^ does not inhibit D1-MSNs in the ventral striatum. As a result, the authors further conclude that prior to using PSAM^4^-GlyR in behavioral studies, validation in the neuron-type is required.

Intracellular chloride levels vary across different neuronal types, and if they are sufficiently elevated so that the chloride reversal potential sits depolarized with respect to the resting membrane potential, activation of a chloride conductance will result in depolarization and sometimes excitation. Given the previous reported electrophysiological properties for D1-MSNs, the primary finding of this paper (PSAM^4^-GlyR activation causes excitation) is not unexpected and thus the conceptual advance seems relatively modest. A more significant contribution could be achieved if validation was conducted in more than one cell type, and in particular with cell types with known differences in chloride homeostasis.

1) In Figure 6 the authors conclude that PSAM^4^-GlyR activation in D1-MSNs reduces GABAergic synaptic inhibition. This conclusion was made by attempting to quantify the reversal potential, but the conductance appears to be almost non-existent for some neurons (e.g. 6C), which makes the calculation for the reversal potential meaningless. The rationale for why the GABA reversal could change makes sense (if prolonged activation of PSAM^4^-GlyR changed the intracellular chloride concentration), although it's not clear why or how activation of PSAM^4^-GlyR could change the conductance of GABA_A_Rs. The amplitudes would be decreased at some membrane potentials (due to changes in driving force), but it’s not clear why they would be reduced at all potentials.

2) N values are relatively low throughout, and/or aren't clearly explained. For example, the N values for experiments in Figure 1, which demonstrated that PSAM^4^-GlyR enhances rather than suppresses c-fos expression in D1-MSNs in vivo, are reported as "6-7 mice/group, but some mice were eliminated because of failure of viral expression". It’s not clear how many mice were eliminated from this already relatively low N value. Also see Figure 4B PSAM for very low n values. For other experiments, it's not clear how many biological replicates were used; e.g. when the number of neurons is reported as 'n', it's not clear how many mice those neurons came from.

3) The Results describe a lot of experimental detail (more appropriate for Materials and methods), but also lack appropriate experimental rationales. For example, the experimental design (Figure 1) involves a fentanyl injection, however there is no explanation for why fentanyl is being used. What is the rationale for this apparently key part of the experimental design?

---

## [Author Response]

[…] The reviewers deemed your work of sufficient interest. We felt that the key finding was the raised c-Fos expression observed in D1 MSNs following activation of PSAM^4^-GlyRs in vivo. However, multiple reviewers raised concerns about the N and the number of mice used not being entirely clear. In particular in Figure 1D the experimental groups have only 5 data points (presumably 5 mice). Considering that the paper is resting on this important finding, the reviewing team asks that you focus your revision on increasing the N for the c-fos experiment to 8-10 animals for the experimental groups to confirm the validity of this key finding.

Per the Editor and Reviewers’ suggestion, we increased the number of animals in the c-fos experiment. The results of the new cohort of mice confirmed the initial finding of increased c-fos expression with PSAM^4^-GlyR activation in saline as well as fentanyl treated mice. Please see updated Figure 1D with individual animals marked as open circles. The current number of mice/group is as follows:

– Control + saline: n = 12 mice.

– PSAM^4^-GlyR + saline: n = 8 mice.

– Control + fentanyl: n = 9 mice.

– PSAM^4^-GlyR + fentanyl: n = 9 mice.

Reviewer #3:Since the initial report of DREADDs (Armbruster 2007, PNAS), chemogenetics has received significant attention for their ability to remotely control neuronal activity. This has included the development of additional chemogenetic tools (proteins and agonists), such as the PSAM/PSEM. The main objective of this study is to validate the use of PSAM^4^-GlyR activated by uPSEM^792^ as an inhibitory chemogenetic tool. By performing extensive electrophysiological characterization, the authors conclude that PSAM^4^-GlyR activated by uPSEM^792^ does not inhibit D1-MSNs in the ventral striatum. As a result, the authors further conclude that prior to using PSAM^4^-GlyR in behavioral studies, validation in the neuron-type is required.Intracellular chloride levels vary across different neuronal types, and if they are sufficiently elevated so that the chloride reversal potential sits depolarized with respect to the resting membrane potential, activation of a chloride conductance will result in depolarization and sometimes excitation. Given the previous reported electrophysiological properties for D1-MSNs, the primary finding of this paper (PSAM^4^-GlyR activation causes excitation) is not unexpected and thus the conceptual advance seems relatively modest. A more significant contribution could be achieved if validation was conducted in more than one cell type, and in particular with cell types with known differences in chloride homeostasis.

We appreciate the Reviewer’s comment and we agree that investigating more than one cell type would offer a substantial contribution as it would catalog neurons for which this particular methodology may be appropriate. However, such a comprehensive study is beyond the scope of this paper. In the original reports by Magnus et al. (2011, 2019), activation of PSAM^4^-GlyR in cortical or hippocampal cells also resulted in an inward current and depolarization of these neurons, but the neuronal silencing was attributed to a shunting effect. We believe the major contribution of our paper is that shunting from chloride channels such as PSAM^4^-GlyR may not be sufficient to silence certain types of neurons and may in fact activate those neurons due to the instability of the chloride gradient. Our results showcase that chloride gradient instability represents a fundamental barrier to using chloride conductances as a reliable strategy to silence neurons.

1) In Figure 6 the authors conclude that PSAM^4^-GlyR activation in D1-MSNs reduces GABAergic synaptic inhibition. This conclusion was made by attempting to quantify the reversal potential, but the conductance appears to be almost non-existent for some neurons (e.g. 6C), which makes the calculation for the reversal potential meaningless. The rationale for why the GABA reversal could change makes sense (if prolonged activation of PSAM^4^-GlyR changed the intracellular chloride concentration), although it's not clear why or how activation of PSAM^4^-GlyR could change the conductance of GABAARs. The amplitudes would be decreased at some membrane potentials (due to changes in driving force), but it’s not clear why they would be reduced at all potentials.

The Reviewer’s point is well taken. We speculate that there are two effects contributing to the reduction in GABA_A_ receptor currents: (1) a change in driving force resulting in a rightward shift in the GABA_A_ receptor reversal potential and (2) a general reduction at all potentials due to postsynaptic electrical shunting. Calculation of GABA_A_ receptor reversal potential is challenging given the small amplitude of the currents. For this reason, rather than attempting to fit individual currents, we performed a linear fit based on the means ± SEM of the five neurons recorded (inset in 6C). We would be happy to remove the results of the reversal potential fit if the Reviewer thinks it is warranted. However, independent of the shift in GABA_A_ receptor reversal potential, the I-V plot in Figure 6C illustrates the dramatically reduced GABA_A_ receptor mediated inhibition upon activation of PSAM^4^-GlyR, which predisposes medium spiny neurons for further excitation. These points are now added to the discussion (page 10):

“The results also show that PSAM^4^-GlyR activation results in ‘apparent cross desensitization’ of GABA_A_ synaptic currents, producing a rightward shift in the GABA_A_ reversal potential and a profound reduction of inhibitory GABA_A_ synaptic currents onto D1-MSNs, likely attributed to electrical shunting since the reduction was observed at all potentials.”

2) N values are relatively low throughout, and/or aren't clearly explained. For example, the N values for experiments in Figure 1, which demonstrated that PSAM^4^-GlyR enhances rather than suppresses c-fos expression in D1-MSNs in vivo, are reported as "6-7 mice/group, but some mice were eliminated because of failure of viral expression". It’s not clear how many mice were eliminated from this already relatively low N value. Also see Figure 4B PSAM for very low n values. For other experiments, it's not clear how many biological replicates were used; e.g. when the number of neurons is reported as 'n', it's not clear how many mice those neurons came from.

The c-Fos experiment (Figure 1) was replicated and the number of mice increased to 812/group (see response to Editor above). This was clarified in the text and the Figure 1 was updated accordingly.

Figure 4B was intended as a qualitative more so than quantitative example of how PSAM^4^-GlyR fails to suppress firing in cell-attached mode and may in fact increase the firing when the membrane is close to firing threshold.

We clarified the number of cells vs. number of mice used in the Materials and methods, Data analysis section (page 13).

3) The Results describe a lot of experimental detail (more appropriate for Materials and methods), but also lack appropriate experimental rationales. For example, the experimental design (Figure 1) involves a fentanyl injection, however there is no explanation for why fentanyl is being used. What is the rationale for this apparently key part of the experimental design?

The rationale for this experiment was based on our initial hypothesis that in vivo activation of PSAM^4^-GlyR will silence firing and lead to a reduction in cFos immunostaining. To avoid a floor-effect where we may fail to observe a decrease in cFos immunostaining by activation of PSAM^4^-GlyR because the level of cFos expression is already low, a cohort of mice was injected with fentanyl since opioids have been shown to increase cFos expression in D1-MSNs. This information is now provided in the Results section (page 3):

“Given the proposed role of PSAM^4^-GlyR activation in silencing neurons (5, 6), we initially hypothesized that activating PSAM^4^-GlyR in vivo will suppress D1-MSNs activity and reduce c-fos expression. To avoid a potential floor effect, we treated mice with fentanyl since opioid exposure has been shown to increase the expression of c-fos and other immediate early genes in striatal neurons including D1-MSNs (14-17).”